# Association between the Nursing Practice Environment and Safety Perception with Patient Safety Culture during COVID-19

**DOI:** 10.3390/ijerph20105909

**Published:** 2023-05-22

**Authors:** Nataly Julissa Membrillo-Pillpe, Jhon Alex Zeladita-Huaman, Kimberlym Jauregui-Soriano, Roberto Zegarra-Chapoñan, Eduardo Franco-Chalco, Gabriela Samillan-Yncio

**Affiliations:** 1Graduate School, Universidad Nacional Mayor de San Marcos, Lima 15001, Peru; 2Academic Department of Nursing, Faculty of Medicine, Universidad Nacional Mayor de San Marcos, Lima 15001, Peru; 3Puesto de Salud Villa Virgen I-2, Cuzco 08720, Peru; 4Faculty of Health Science, Universidad María Auxiliadora, Lima 15408, Peru; 5Research Department, Universidad María Auxiliadora, Lima 15408, Peru

**Keywords:** patient safety, environment, leadership, organizational culture, nurse clinicians

## Abstract

Aims: In this study, we aimed to identify the relationship between nursing practice environments (NPEs) and safety perceptions with patient safety culture (PSC) during COVID-19. Design: We conducted a quantitative, non-experimental, correlational, and cross-sectional study. We interviewed 211 nurses from Peru using two scales: PES–NWI and HSOPSC. We used the Shapiro–Wilk test and Spearman’s coefficient and estimated two regression models. Results: NPE was reported as favorable by 45.5% of the participants, and PSC was reported as neutral by 61.1%. Safety perception, the workplace, and NPE predict PSC. All NPE factors were correlated with PSC. However, safety perception, support of nurses subscale, the nurse manager’s ability, and leadership were predictors of PSC. Conclusion: To promote a safe work culture, health institutions should foster leadership that prioritizes safety, strengthens managers’ abilities, encourages interprofessional collaboration, and considers nurses’ feedback for constant improvement.

## 1. Introduction

The COVID-19 pandemic drastically changed health professionals’ day-to-day work and interdisciplinary dynamics in health facilities [1]. As a result of this viral disease, different health systems collapsed as healthcare demand increased. Health staff handled this adverse situation to ensure timely care and prevent new infections [2]. In this context, health workers faced the sudden change of being assigned new services, functions, and preventative activities. The fatigue and pressure of these new responsibilities were overcome by the synergy of professional teamwork and complementary training [3].

During COVID-19, the workplace environment of healthcare professionals was characterized by exposure to emotionally exhausting situations, high-pressure interprofessional interactions, risky working conditions, long working hours, and poor remuneration. Nurses experienced greater fear, concern, fatalism [4], stress, fatigue, and infection-related anxiety due to the increased workload, staff shortage, and insufficient distribution and supply of personal protective equipment [5].

Since the book To Err Is Human: Building a Safer Health System was published, healthcare professionals were encouraged to prioritize patient safety and reduce adverse events [6], as unsafe healthcare is one of the leading causes of death and disability worldwide [7]. In addition, hospitals in low- and middle-income countries have reported 134 million adverse events annually due to unsafe healthcare, resulting in 2.6 million deaths worldwide [8].

Researchers use different methods to measure patient safety. Some use a series of questions that determine patient safety implementation [9], such as safety perception [10,11] or patient safety outcomes in terms of adverse event reporting [12,13]. However, patient safety culture (PSC) is used in different countries to assess patient safety [14,15]. PSC is the culmination of health staff’s values, attitudes, perceptions, competence, and behavioral patterns in health systems. It identifies risks, foresees safety issues, fosters a healthy work setting, favors multidisciplinary teamwork, focuses on finding solutions, allocates necessary resources, implements management strategies, and assesses changes [16]. In this regard, PSC—often perceived as implicit—refers to a set of interactive elements implying certain ways of doing, thinking, and administrating [17].

Before the COVID-19 pandemic, nurses in South Korea [18] and the United States [19] reported having an excellent level of PSC. However, one patient per nurse was associated with a 2% increase in the likelihood of rating patient safety as poor [18]. A study conducted in Peru reported a medium level of patient safety in hospitals [20]. Another study highlighted that patient safety perception varies according to the type of health facility. The percentage of public hospital workers who rated patient safety as excellent (35%) was higher than public sector workers [21].

Subsequently, studies conducted during the pandemic show that nurses perceived moderately unfavorable PSC [22,23], especially in units caring for larger numbers of patients, such as internal medicine, intensive care units, surgery, and pediatrics [12]. Patient safety was rated low for nurses who, during their last shift, reported caring for a greater number of patients than usual and poorer working conditions [10]. Considering the pandemic’s impact on PSC [22], analyzing PSC in different working environments and units is crucial.

Previous studies highlighted that patient safety implementation is associated with the nursing staff’s educational level, professional experience, training in patient safety [9], and the work environment [24]. Likewise, long shifts, several night shifts, overtime hours, fewer days off [19], and an increased number of patients per nurse [18] were associated with low PSC. By contrast, leadership [25], high experience level, and university hospital employment were associated with greater PSC perception [26]. In addition, nurses who were aware of their supervisors’ or managers’ expectations of them, received support and feedback, improved upon mistakes, and worked as a team reported adverse events more frequently [12,26].

Furthermore, the nursing practice environment (NPE) refers to a work setting’s organizational characteristics that can facilitate or constrain professional nursing practice. Nursing professionals’ positions in a healthcare organization and interactions between nurses, hospital directors, and doctors are examples of NPEs [27]. Studying NPEs is important because fostering healthier work environments adds to nurses’ satisfaction [28], optimizes job performance, improves the quality of patient care [29], increases professionals’ and patients’ well-being [30], and favors adverse event reporting [31].

Cross-sectional studies agree that NPE predicts PSC [15,19,32]. Moreover, a longitudinal study demonstrated that a work setting that addresses patients’ needs and favors their staff’s commitment is essential for a long-term safety environment [33]. However, a few studies that analyzed the predictive value of NPE subscales and PSC showed mixed results [14,34]. Therefore, we aimed to identify the association between NPE and safety perception with PSC during COVID-19.

## 2. Materials and Methods

### 2.1. Design and Settings

This quantitative, non-experimental, correlational, and cross-sectional study was conducted during the second wave of the COVID-19 pandemic. We required informed consent from all participants; those who provided consent had access to the questionnaire. The participants were nurses working in two city hospitals (Lima and Callao), where most of the health services in Peru are concentrated. The research project was approved by the Institutional Research Ethics Committee of the Universidad Norbert Wiener (File 352-2021). Eighty-six cases of nurses working in critical areas were included in a Master’s Degree thesis for Teaching and Research in Health from the Universidad Nacional Mayor de San Marcos, titled “Association of the nursing practice environment with patient safety culture in nurses who work in critical areas, 2020”.

For the most part, health systems in Latin America are fragmented and segmented. Universalizing healthcare in the region is still in progress, and even today, there are gaps between the public and private sectors. Concerning healthcare provision, the public sector works under subsidized schemes (Integrated Health System or SIS in Peru) and direct, contributory schemes (social security). Similarly, the private sector works under one contributory scheme (healthcare providers and specialized clinics, among others) [21].

### 2.2. Sample and Procedures

Using G*Power 3.1.9.7 software (Dusseldorf University, Dusseldorf, Germany), we determined that the sample should have at least 43 participants to achieve 80% statistical power with a 5% probability of committing a type I error. We also estimated minor effects for the present study. However, we decided to obtain a larger sample size for a higher level of representativeness. Thus, the study population consisted of 211 nurses working in primary healthcare facilities, public hospitals, private clinics, and military hospitals. We used non-probability snowball sampling.

Prior to data collection, the study was approved by an ethics committee. We applied the survey technique using an online self-administered questionnaire in Google Forms. We addressed the form to nurses pursuing postgraduate studies and distributed it via hospitals’ and universities’ social networks (WhatsApp, Facebook, and Instagram). Professional practice and academic activities were not interrupted at any moment.

### 2.3. Measurement Tools

We used the 31-item Practice Environment Scale of the Nursing Work Index (PES–NWI) to measure NPE. This scale has five subscales: (1) nurse participation in hospital affairs; (2) nursing foundations for quality of care; (3) nurse manager ability, leadership, and support of nurses; (4) staffing and resource adequacy; and (5) collegial nurse–physician relations. This Likert scale has four response options ranging from 1 (strongly disagree) to 4 (strongly agree) and reports adequate reliability (Cronbach’s alpha greater than 0.80). Its final value is considered “favorable” if the mean scores of four or five subscales are >2.5, mixed if the means of two or three subscales are >2.5, or “unfavorable” if one or none of the five subscales reach a mean score of 2.5 [27]. This scale was validated in the Hispanic population with adequate internal consistency, as determined by Cronbach’s alpha coefficient of 0.89 [35]. Additionally, it is highly reliable based on the data collected (McDonald’s omega coefficient = 0.95).

Regarding PSC measurement, we used the Hospital Survey on Patient Safety Culture version 2.0 developed by the Agency for Healthcare Research and Quality [36]. This version differs from the older one and has 32 items grouped into 10 composite dimensions in sets of two or more survey items: (1) teamwork; (2) staffing and work pace; (3) organizational learning—continuous improvement; (4) response to error; (5) supervisor, manager, or clinical leader support for patient safety; (6) communication about error; (7) communication openness; (8) reporting patient safety events; (9) hospital management support for patient safety; (10) handoffs and information exchange. It also has six demographic items and two on error reporting and safety culture. It uses a five-point Likert scale (strongly disagree to strongly agree) or frequency scales (never to always) and includes positively and negatively worded items. Regarding data analysis, to calculate the percentage obtained for each dimension, it is necessary to average the percentage of positive scores for each item included in the dimension. We should note that disagreeing with a negatively worded item indicates a positive response. According to the Agency for Healthcare Research and Quality [36], the PSC level was designed as positive, neutral, and negative. Averaging the percentage of positive scores at the item level results in the average percentage of that dimension: If it is <50%, it is considered negative; if it remains in the range of 50–60%, it is considered neutral; if it is ≥70%, it is positive. This scale was adapted to and validated by a cross-cultural study of the Peruvian population [37]. In addition, it has high reliability according to the data collected (McDonald’s omega coefficient = 0.94).

Safety perception was measured based on answers to the following question: How would you rate patient safety in your service/work unit? The response alternatives were poor, fair, good, very good, and excellent.

As proposed by AHRQ, the worst possible score for the item was placed first and the best possible score last to ensure that the participants read all the answer options.

We also inquired about their gender, age, place of work, working hours, length of service, whether their work was performed in critical areas (emergency or intensive care units), and whether their job entailed contact with patients diagnosed with COVID-19.

### 2.4. Analysis

To meet the objectives of our study, we first analyzed the variables’ descriptive statistics. We reviewed the normality assumption of the continuous variables with the Shapiro–Wilk test to confirm the use of parametric tests. Subsequently, given the absence of normality in the work environment and safety culture variables, we used Spearman’s coefficient to estimate the correlations between the work environment subscales and safety culture dimensions. Finally, we estimated two regression models to predict safety culture. To that end, we used a stepwise variable selection algorithm to determine predictor variables based on changes in Akaike’s information criteria. These variables had the best predictive capacity for the response variable. For the first model, we included the work environment subscales within the predictor variables and other sociodemographic data. For the second model, we included the entire scale within the predictor variables but removed the dimensions. We transformed the safety culture variable using Tukey’s ladder of power since it does not follow a normal distribution [38]. This step allowed us to bring the data distribution closer to a more normal distribution. The resulting models underwent a diagnosis of outliers and assumptions. We observed no outlier cases and proved all multiple regression assumptions. As for interpreting regression parameters, we calculated standardized coefficients. We conducted all analyses using R software, version 4.2.1 (R Core Team, Auckland, New Zeland), [39].

## 3. Results

The questionnaire was completed by 211 nurses, 193 of whom were female (91.5%). Most participants were aged between 25 and 44 (79.6%). Regarding their workplace, 114 (54.0%) worked in hospitals and 53 (25.1%) in primary healthcare facilities. As for the length of service, 55 (26.1%) had <1 year of service, 61 (28.9%) had 1–4 years of service, and 45 (21.3%) had 5–9 years of service. Table 1 shows other characteristics related to the participants’ working conditions.

### 3.1. Nursing Practice Environment

NPE was favorable for 45.5% of respondents, neutral for 31.2%, and unfavorable for 23.2%. 

Table 2 shows the means and standard deviations of the NPE subscales, the PSC scale, and its dimensions. It also shows the normality test results of the study variables. None of the variables follow a normal distribution because the skewness and kurtosis scores differ from 0. Similarly, the significance value of the Shapiro–Wilk test in all cases is <0.001, implying that we cannot assume normality in any of the cases.

### 3.2. Patient Safety Culture

Regarding the PSC level, 61.1% of nursing professionals indicated that it was neutral, 31.8% positive, and 7.1% negative. 

Table 3 shows the bivariate relationships between NPE and PSC, as determined by Spearman’s correlation coefficient. All correlations are statistically significant when *p* < 0.001. The NPE subscales correlate positively and strongly between the same subscales and the total scale (0.46 < rho < 0.84), which was expected. Furthermore, the first NPE subscale measuring nurse participation in hospital affairs correlates positively and strongly with PSC (rho = 0.466). Similarly, the second NPE subscale measuring nursing foundations for quality of care also correlates strongly and positively with PSC (rho = 0.500). The third NPE subscale measuring nurse manager ability, leadership, and support of nurses has a strong and positive relationship with PSC (rho = 0.497). Similarly, the fourth NPE subscale measuring staffing and resource adequacy correlates positively and strongly with PSC (rho = 0.409). The fifth NPE subscale measuring collegial nurse–physician relations correlates strongly and positively with PSC (rho = 0.430). Finally, the total NPE scale strongly and positively correlates with PSC (rho = 0.570). 

Table 4 shows the multiple regression model. The algorithm-selected variables that contributed the most to predicting safety culture were the perception of patient safety during service and three NPE subscales. However, only the perception of patient safety during service and the nurse manager’s ability, leadership, and support of nurses subscale had significant effects. Specifically, nurses who reported perceiving good, very good, and excellent patient safety during service also reported higher PSC scores than those who reported poor patient safety. These differences are moderate to strong according to the standardized regression coefficients (B = 0.36, B = 0.52, B = 0.32, respectively). Moreover, the higher the participants rated the nurse manager’s ability, leadership, and support of nurses subscale, the better PSC in their work environment. According to the standardized coefficient, this is a moderate effect (B = 0.23). This model explains 50% of safety culture variance according to the R^2^ indicator.

Table 5 shows the multiple regression model selected to predict PSC. The algorithm-selected variables that contributed the most to predicting PSC were work environment, perception of patient safety during service, and workplace. Moreover, all these variables had a statistically significant effect. The higher the participants rated the total work environment scale, the better they reported PSC. This effect is strong according to the standardized regression coefficient (B = 0.40). Additionally, nurses who reported very good and excellent patient safety during service also reported higher PSC scores than those who reported poor patient safety. According to standardized regression coefficients, these differences are moderate to strong (B = 0.47, B = 0.31, respectively). Finally, nurses working in military hospitals reported higher PSC levels than nurses in other hospitals. This difference is slight according to the standardized regression coefficient (B = 0.11). This model explains 51% of the PSC variance according to the R^2^ indicator.

## 4. Discussion

The main finding of this study, conducted during the second wave of the COVID-19 pandemic, is that safety perception, workplace, and NPE predict the PSC score. In other words, safety culture in a healthcare facility is determined by the working conditions of the environment, the facility’s characteristics, as well as safety perception. This finding highlights the impact of nurses’ perceptions of working conditions on patient safety. This aspect is relevant, considering a healthy work setting could reduce patient deaths and failure-to-rescue by approximately 2 and 3%, even in hospitals with poor nurse staffing [29].

In this study, NPE’s predictive role in PSC is consistent with studies of nurses in critical care units in Oman [32], hospitals in Saudi Arabia [15], and health institutions providing pediatric care in the United States [19]. Similarly, work environments reportedly predict the degree of patient safety [11]. In addition, a systematic review highlighted that healthy work environments were significantly related to hospital safety climate and culture [29]. A longitudinal study conducted with Norwegian health staff reported that a work environment favoring safe incident reporting, innovation, and teamwork led to positive changes in the safety climate [33]. However, systematic reviews of primary studies measuring the relationship between work environment characteristics and patient safety outcomes, such as medication errors, overall patient safety, and nosocomial infections, reported contradictory results [40,41].

Regarding our bivariate analysis of the relationship between NPE subscales and PSC, we found that the scores of all subscales were correlated with the PSC score. This finding coincides with studies of nurses in Jordan [14], Saudi Arabia [15], and Poland [10]. However, our regression analysis revealed the explanatory value of the nurse manager’s ability, leadership, and support of nurses subscale in PSC. This finding can be explained by the leader–member exchange theory. One study using this model found that the contact frequency between leaders and members favors good relationships, thus improving PSC [25]. Another study reported that managers with good interpersonal skills can influence workplace safety climate [42]. In addition, this finding is similar to scientific evidence associating PSC with managers’ expectations and support in safety management [16]. Other studies also highlight that management expectations, safety actions [12], and interprofessional collaboration [13] explain patient safety outcomes such as adverse event reporting.

By contrast, a study conducted in Turkey found that PSC is predicted by three other subscales of NPE—nurse participation in hospital affairs, nursing foundations for quality of care, and collegial nurse–physician relations [34]. Another study in Jordan indicated that nurses who perceived poor staffing, resource adequacy, and negative collegial nurse–physician relations were likelier to report a lower perception of PSC [14]. One explanation for the discrepancy between our results and those of said studies might be that safety perception is strongly related to PSC and the NPE subscales. Therefore, this provides a strong statistical control effect that leaves little variability explained by the NPE subscales. Furthermore, in this study, staffing does not explain PSC variability, which could result from the association between staffing and patient safety perception [43,44].

Considering the number of millennials entering the nursing workforce daily, nurse leaders must create development opportunities by eliminating generational barriers and providing meaningful mentorship that encourages understanding and a positive attitude at work [45]. To promote health reforms in nursing, nurse leaders should fulfill leadership competencies in the four domains (professional, clinical, health policy, and health systems) highlighted by an integrative review study [46].

Another PSC predictor reported in the study is the perception of safety. One study reported an association between patient safety and the work environment [9]. Another suggested that safety perception is related to PSC [16]. 

In addition, we found that nurses in military hospitals scored a higher PSC than those in public hospitals. By contrast, a study with Peruvian health professionals reported no statistically significant difference in the percentage of participants from both health facility types who perceived PSC as excellent or very good. However, the same study highlighted a gap in PSC perception between public and private hospital workers [21]. The composition of our study sample might explain this difference. Another study in Oman reported that nurses in university hospitals had a higher PSC perception [26]. Conversely, a study in Venezuela reported no differences in PSC between public and private hospitals [12]. Consequently, studies in diverse contexts should be conducted to explore this association. Although the evidence suggests differences in PSC perception among health facility types, there is no consensus on its predictive value on PSC.

The neutral PSC level reported in this study aligns with studies with nurses from Latin American countries such as Brazil [23], Venezuela [12], and Peru [21], who reported low PSC levels during the COVID-19 pandemic. It is also consistent with another study in Poland, where half of the nurses perceived an intermediate level of patient safety in their work environment [10]. However, these levels differ from studies conducted before the pandemic [18,19].

Therefore, nurses are not in favor or against it since their answers were “neither agree nor disagree” [47]. On the other hand, if one considers pre-pandemic studies, Peruvian nurses reported a medium [20] or positive [48] CSP level. This decline is likely due to nurses’ poor working conditions during the COVID-19 pandemic. A study comparing PSC in Portugal before and during the COVID-19 pandemic found that the percentage of positive responses varied in nine out of twelve dimensions. Specifically, it reported an increase in five indicators and a decrease in four dimensions, mainly regarding the overall perception of patient safety, frequency of adverse reports, and staffing [22].

The favorable NPE level reported in this study reveals that most participants perceive good working conditions in Peruvian hospitals. This finding is consistent with pre-pandemic studies with nurses in Ireland [31], Jordan [14], and Poland [10]. However, it is inconsistent with a study from Turkey, where nurses were found to have poor working conditions [34]. These disparities may result from different working conditions among countries. It is also worth mentioning that a study in Portugal comparing NPE before and during the COVID-19 pandemic identified an increase in indicators such as outcome and structural components but a decrease in indicators such as collaboration and teamwork, planning, assessment, and continuity of care—all from the processing component [22].

Our findings have important implications for hospital management practices because of the proven association between nurses’ working conditions and PSC. We determined that management skills, leadership, and interprofessional collaboration explain the variability in PSC. Therefore, key aspects include promoting a work environment where management and leadership are centered on patient safety and interprofessional collaboration. In this sense, Peruvian nurses who assume leadership in caring (department heads and supervisors) and teaching (directors and higher education authorities) must coordinate strategies at the microsystem level with multidisciplinary teams. They must encourage continuous improvement in undergraduate and postgraduate training so that they develop internationally defined leadership competencies, self-confidence, and professional autonomy. In this way, an adequate work environment is promoted and patients’ safety is guaranteed.

This study has some limitations. On the one hand, these findings can only be described as predictors, given that this was a correlational and cross-sectional study with multivariate analysis. However, the proposed model explains 50% of the total variance. On the other hand, the data were collected using a self-administered online form so that participants were not influenced by social desirability bias. However, prior to collection, the research team explained the purpose and confidentiality of the study to the participants.

## 5. Conclusions

This study confirms that safety perception, workplace, and NPE predict the PSC score in a predictive model, explaining 51% of the variability. The relationship between NPE subscales and PSC is debatable; however, this predictive model aligns with previous studies by demonstrating the correlation of all NPE subscales with PSC. The study, however, differs from previous findings by reporting that safety perception and only one of the NPE subscales (nurse manager’s ability, leadership, and support of nurses) are predictors of PSC. Peruvian nurses present an uncertain level of PSC but perceive favorable NPE.

Finally, to promote a safe work culture in health institutions in low- and middle-income countries generally characterized by understaffing, we suggest fostering leadership among the nursing team. This leadership should prioritize patient safety, strengthen management skills, encourage interprofessional collaboration, and consider nurses’ opinions and safety reports for continuous improvement.

## Figures and Tables

**Table 1 ijerph-20-05909-t001:** Work-related characteristics of the sample (*N* = 211).

	Number	%
Workplace		
Hospital	114	54.0
Clinic	36	17.1
Health center	53	25.1
Military hospital	5	2.4
Other private healthcare facility	3	1.4
Length of service (years)		
<1	55	26.1
1–4	61	28.9
5–9	45	21.3
10–14	17	8.1
>15	33	15.6
Working hours		
<20	19	9.0
20–29	36	17.1
30–39	103	48.8
>40	53	25.1
Working in critical area		
Yes	85	40.3
No	126	59.7
Patient-safety-related events		
None	60	28.4
1–2	87	41.2
3–5	43	20.4
6–9	10	4.7
>10	11	5.3
Perception of patient safety during service		
Poor	6	2.9
Fair	64	30.3
Good	87	41.2
Very good	49	23.2
Excellent	5	2.4
Contact with COVID-19 patients		
Yes	196	92.9
No	15	7.1

**Table 2 ijerph-20-05909-t002:** Descriptive statistics and normality tests for the subscales and total scale of nursing practice environments and safety culture (*N* = 211).

Variable	Mean	Standard Deviation	Skewness	Kurtosis	Shapiro–Wilk	*p* Shapiro–Wilk
Subscale: nurse participation in hospital affairs	2.76	0.47	−0.76	1.19	0.95	<0.001
Subscale: nursing foundations for quality of care	2.71	0.37	−0.79	0.67	0.95	<0.001
Subscale: nurse manager’s ability, leadership, and support of nurses	2.58	0.64	−0.39	0.14	0.97	<0.001
Subscale: staffing and resource adequacy	2.27	0.55	−0.33	−0.15	0.96	<0.001
Subscale: collegial nurse–physician relations	2.64	0.49	−0.29	0.61	0.94	<0.001
Nursing practice environment safety culture	2.64	0.39	−0.68	0.58	0.97	<0.001

**Table 3 ijerph-20-05909-t003:** Spearman’s correlation matrix for the subscales and total scale of nursing practice environments and safety culture (*N* = 211).

	1	2	3	4	5	6
1. Subscale: nurse participation in hospital affairs						
2. Subscale: nursing foundations for quality of care	0.532 *					
3. Subscale: nurse manager’s ability, leadership, and support of nurses	0.656 *	0.591 *				
4. Subscale: staffing and resource adequacy	0.494 *	0.544 *	0.560 *			
5. Subscale: collegial nurse–physician relations	0.526 *	0.533 *	0.526 *	0.467 *		
6. Nursing practice environments	0.830 *	0.823 *	0.844 *	0.733 *	0.690 *	
7. Safety culture	0.466 *	0.500 *	0.497 *	0.409 *	0.430 *	0.570 *

* *p* < 0.001.

**Table 4 ijerph-20-05909-t004:** Regression model for predicting safety culture based on the subscales of nursing practice environments (*N* = 211).

	b	SE	B	*p*
Intercept	−0.93	0.01		<0.001
Perception of safety (fair)	0.01	0.01	0.15	0.32
Perception of safety (good)	0.01	0.01	0.36	0.02
Perception of safety (very good)	0.02	0.01	0.52	<0.001
Perception of safety (excellent)	0.04	0.01	0.32	<0.001
Subscale: nurse manager’s ability, leadership, and support of nurses	0.01	0.00	0.23	<0.001
Subscale: nursing foundations for quality of care	0.01	0.00	0.14	0.05
Subscale: collegial nurse–physician relations	0.01	0.00	0.11	0.08
R^2^	0.50			

Note: The reference category for perception of safety is poor. b = unstandardized coefficient; SE = standard error; B = standardized coefficient; *p* = *p*-value.

**Table 5 ijerph-20-05909-t005:** Regression model predicting safety culture based on the total scale of nursing practice environments (*N* = 211).

	b	SE	B	*p*
Intercept	−0.93	0.01		<0.001
Nursing practice environments	0.02	0.00	0.40	<0.001
Perception of safety (fair)	0.00	0.01	0.08	0.57
Perception of safety (good)	0.01	0.01	0.29	0.07
Perception of safety (very good)	0.02	0.01	0.47	<0.001
Perception of safety (excellent)	0.04	0.01	0.31	<0.001
Workplace (clinic)	0.00	0.00	0.08	0.13
Workplace (health center)	0.00	0.00	0.05	0.35
Workplace (military hospital)	0.01	0.01	0.11	0.03
Workplace (other healthcare facility)	−0.01	0.01	−0.05	0.34
R^2^	0.51			

Note: The reference category for perception of safety is poor. The reference category for workplace is hospital. b = unstandardized coefficient; SE = standard error; B = standardized coefficient *p* = *p*-value.

## Data Availability

The datasets generated or analyzed during this study are available from the corresponding author on reasonable request.

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
