# Peer review of "Association between the Nursing Practice Environment and Safety Perception with Patient Safety Culture during COVID-19"

_ijerph, 2023, doi:10.3390/ijerph20105909_

Round 1

Reviewer 1 Report

Roberto Zegarra-Chapoñan et al. submitted to IJERPH an article focusing to the relationship of nursing practice environment and safety perception with patient safety culture during COVID-19 pandemic.

This manuscript is well structured and it is intended for experts in this field, it lends itself to a fluid reading, being supported by a suitable bibliography. The sampling method, the statistical analyzes and the limits of the study are well explained.

Here are my other suggestions:

- LL 30-32: in corroborating this statement, please also evaluate the contents of this recent study: DOI: 10.3390/healthcare10101906

- LL 246-247: please provide the correct caption of the table

- in dealing with nursing leadership, please consider the results and discussions that emerge from the following works: DOI: 10.1111/jan.14092 - DOI: 10.1097/01.NURSE.0000580656.81188.ee

- are there Nursing Managers in your organizational context? How concretely can they contribute to promoting actions aimed at improving nurses' working conditions?

- I strongly suggest an English language check, both for the style used and the syntax, performed by at least a native English speaking colleague. 

Thank you for your efforts in perfecting this important article!

Author Response

We are grateful to you and the reviewers for your time in helping us to advance with this opportunity for sharing the findings of our study. We appreciate the comments and suggestions that we have received. We have addressed them to the best of our ability as noted below.

Furthermore, our manuscript was submitted to English editing, as suggested.

We thank you for your continued interest in our research.

Reviewer: 1

Comments to the Author

Roberto Zegarra-Chapoñan et al. submitted to IJERPH an article focusing to the relationship of nursing practice environment and safety perception with patient safety culture during COVID-19 pandemic. This manuscript is well structured and it is intended for experts in this field, it lends itself to a fluid reading, being supported by a suitable bibliography. The sampling method, the statistical analyzes and the limits of the study are well explained. Here are my other suggestions:

Comment 1:

- LL 30-32: in corroborating this statement, please also evaluate the contents of this recent study: DOI: 10.3390/healthcare10101906

Response 1:

Thank you very much for your comment and suggestion. We have considered your suggestion appropriate and so it was incorporated in at the end of the first paragraph of the introduction

Comment 2:

- LL 246-247: please provide the correct caption of the table

Response 2: We have added the correct caption of the table.

Comment 3:

In dealing with nursing leadership, please consider the results and discussions that emerge from the following works: DOI: 10.1111/jan.14092 - DOI: 10.1097/01.NURSE.0000580656.81188.ee

Response 3: We have considered your suggestion appropriate and therefore both were incorporated as the fifth paragraph of the discussion.

Comment 4:

Are there Nursing Managers in your organizational context? How concretely can they contribute to promoting actions aimed at improving nurses' working conditions?

Response 4: We have added the following paragraph to the discussion (line 347 – 353).

In this sense, Peruvian nurses who assume leadership in caring (department heads and supervisors) and teaching (directors and higher education authorities) must coordinate strategies at the microsystem level with multidisciplinary teams. They must encourage continuous improvement in undergraduate and postgraduate training so that they develop internationally defined leadership competencies, self-confidence and professional autonomy. In this way, an adequate work environment is promoted and patients’ safety is guaranteed.

Comment 5:

- I strongly suggest an English language check, both for the style used and the syntax, performed by at least a native English speaking colleague. 

Thank you for your efforts in perfecting this important article!

Response 5: Once the observations were incorporated, the new version was sent to MDPI for English editing. A certificate issued by Author Services is attached.

Reviewer 2 Report

Dear Authors,

I have some concerns regarding the first part of your article (1. Introduction), where it is clear that extensive editing of English language and style is mandatory.

Check Table 5, there is a standard description, not the one intended, for sure.

As for the design, it would have been interesting to have results before the pandemic, for a case-control study...

Author Response

We are grateful to you and the reviewers for your time in helping us to advance with this opportunity for sharing the findings of our study. We appreciate the comments and suggestions that we have received. We have addressed them to the best of our ability as noted below.

Furthermore, our manuscript was submitted to English editing, as suggested.

We thank you for your continued interest in our research.

Comments to the Author

Comment 1:

Dear Authors, I have some concerns regarding the first part of your article (1. Introduction), where it is clear that extensive editing of English language and style is mandatory.

Response 1: Thank you very much for your suggestions. The revised version was submitted to MDPI for English editing. Certificate issued by Author Services is attached.

Comment 2:

Check Table 5, there is a standard description, not the one intended, for sure.

Response 2: We have added the correct caption of the table.

Comment 3:

As for the design, it would have been interesting to have results before the pandemic, for a case-control study...

Response 3: We appreciate the observation and agree that a pre-pandemic control group would have provided valuable information about the phenomenon under study. Unfortunately, the conceptualization of the study was done after the onset of the pandemic, so data could not be collected in the requested season.

Reviewer 3 Report

This is valuable imput in Patient Safety Culture in extreem challenge of scrutinity of beds and staff during pandemics. It is also a unique experience. Technically is is a fair analysis and confirmes the results obtained by other authors however I miss some information ;

Was poor perceptionof patiets safety in most busy deparetments? When was the survey  performed or what period of pandemic was assessed. And the excellent patients safety -was it outpatient clinic_? were participants instructed to chose the worst possoble  score for item ? I think this kind of information is useful in analysis like this.

Author Response

We are grateful to you and the reviewers for your time in helping us to advance with this opportunity for sharing the findings of our study. We appreciate the comments and suggestions that we have received. We have addressed them to the best of our ability as noted below.

We thank you for your continued interest in our research.

Reviewer: 3

Comments to the Authors

This is valuable input in Patient Safety Culture in extreme challenge of scrutiny of beds and staff during pandemics. It is also a unique experience. Technically is a fair analysis and confirms the results obtained by other authors however I miss some information:

Comment 1:

Was poor perception of patients safety in most busy departments? And the excellent patients safety - was it outpatient clinic?

Response 1: Thank you very much for your suggestions. We have restructured the third to fifth paragraphs of the introduction to address factors associated with the perception of patient safety both before and during the COVID-19 pandemic.

Comment 2:

When was the survey performed or what period of pandemic was assessed.

Response 2: We have added in the 2.1 Design and Settings section that “study was conducted during the second wave of the COVID-19 pandemic” (Line 96 – 97).

Comment 3:

Were participants instructed to choose the worst possible score for item? I think this kind of information is useful in analysis like this.

Response 3: We have added in the 2.3 Measurement Tools section that “the worst possible score for the item was placed first and the best possible score last to ensure that the participants read all the answer options” (Line 160 – 161)

Reviewer 4 Report

This research presents an interesting topic that connects the nursing practice environment and patient safety culture in a critical time of COVID-19. This article seems fitting the special issue.

Still a number of concerns are stated as follows before considering publishing:

In the introduction, the review of PSC is too general with an absence of proper literature review section. For example, in the ending sentence of the paragraph, it is stated that PSC is a ‘tactic’. How is this tactic related to COVID-19 situation? What is the implications?

In section 2.3, the description for PSC is not sufficient. While the different dimensions of NPE are explained in detail, the PSC did not cover the same level of detail. It is also observed that the results for the PSC measurement it not sufficiently discussed. Instead, perception of safety was measured based on a particular question, “how would you rate patient safety?”.

For the question mentioned above, it is unsure if the scale “poor, fair, good, very good, excellent” is proper.

In section 3.2, how should we interpret a results with 31.8 positive, 61.1 uncertain and 7.1 negative? Is it actually a low level of PSC as reported in line 314-315?

It is not sure if the multiple regression model is correctly constructed. Is all factors mentioned above were independent variables or the perception of safety considered as dependent variable and the NPE attributes are as independent variables? Given a lot of comparison of scores were conducted rather than to discuss about the relationship of scores, should other statistical methods e.g. ANOVA, be considered?

Author Response

Dear Mrs. Dunja Stojanac
Assistant Editor/ MDPI Belgrade

We are grateful to you and the reviewers for your time in helping us to advance with this opportunity for sharing the findings of our study. We appreciate the comments and suggestions that we have received. We have addressed them to the best of our ability as noted below.

We thank you for your continued interest in our research.

Reviewer: 4

Comments to the Authors

This research presents an interesting topic that connects the nursing practice environment and patient safety culture in a critical time of COVID-19. This article seems fitting the special issue. Still a number of concerns are stated as follows before considering publishing:

Comment 1:

In the introduction, the review of PSC is too general with an absence of proper literature review section. For example, in the ending sentence of the paragraph, it is stated that PSC is a ‘tactic’. How is this tactic related to COVID-19 situation? What is the implications?

Response 1: Thank you very much for your suggestions. The introduction (third to fifth paragraph) has been restructured to describe the state of the art of patient safety perception both before and during the COVID-19 pandemic. Likewise, the term that was recorded was “tacit”, but since it can be ambiguous, it was changed to “implicit”.

Comment 2:

In section 2.3, the description for PSC is not sufficient. While the different dimensions of NPE are explained in detail, the PSC did not cover the same level of detail.

Response 2: We have made the requested changes in section 2.3. (Line 140 -144)

Comment 3:

It is also observed that the results for the PSC measurement it not sufficiently discussed.

Response 3: We have added the following paragraph to the discussion about of the neutral level of PSC measured. (Line 325 – 322)

Therefore, nurses are not in favor or against it since their answers were “neither agree nor disagree” [47]. On the other hand, if one considers pre-pandemic studies, Peruvian nurses reported a medium [20] or positive [48] CSP level. This decline is likely due to nurses’ poor working conditions during the COVID-19 pandemic. A study comparing PSC in Portugal before and during the COVID-19 pandemic found that the percentage of positive responses varied in nine out of twelve dimensions. Specifically, it reported an in-crease in five indicators and a decrease in four dimensions, mainly regarding the overall perception of patient safety, frequency of adverse reports, and staffing [22].

Comment 4:

Instead, perception of safety was measured based on a particular question, “how would you rate patient safety?”. For the question mentioned above, it is unsure if the scale “poor, fair, good, very good, excellent” is proper.

Response 4:

It is correct, we included a question for the participant to qualify from their perception of patient safety in their service, whose alternatives were: poor, fair, good, very good, excellent. In this study, this question, which was considered as a possible predictor because it was reported by other studies, after the statistical analysis, it was found that it is indeed a predictor of PSC.

Comment 5:

In section 3.2, How should we interpret a result with 31.8 positive, 61.1 uncertain and 7.1 negative? Is it actually a low level of PSC as reported in line 314-315?

Response 5:

Thank you very much for identifying the error in reporting the PSC level, since the "uncertain" level, which we had entered, was changed to a "neutral" level. In the study we found that the neutral level predominated, which was considered in the discussion.

Comment 6:

 It is not sure if the multiple regression model is correctly constructed. Is all factors mentioned above were independent variables or the perception of safety considered as dependent variable and the NPE attributes are as independent variables? Given a lot of comparison of scores were conducted rather than to discuss about the relationship of scores, should other statistical methods e.g. ANOVA, be considered?

Response 6: We appreciate your comments and suggestions regarding the choice of analysis method for our study. We understand that you have proposed the use of ANOVA as an alternative to the linear regression model. However, we would like to provide a more detailed rationale for our choice to use a linear regression model in this context.  First, our study focuses on analyzing the individual effect of multiple independent variables, both categorical and continuous, on the dependent variable. Although ANOVA is a robust and useful method for comparing the means of different groups, its primary application is to determine whether there is a significant difference between group means. On the other hand, the linear regression model allows us to estimate the individual effect of each independent variable on the dependent variable, which is more in line with our research objectives.  As for the handling of categorical variables, it is possible to include them in a linear regression model through the technique of coding Dummy variables, the mode under which they were entered in the model used. This approach allows us to examine the individual effect of each category of a categorical variable on the dependent variable, and facilitates the interpretation of the regression coefficients in relation to the reference category.  Finally, although both methods can help us to identify significant relationships between variables, the linear regression model offers greater ease of interpretation and communication of the results, especially when a considerable number of independent variables are involved. In this sense, we believe that linear regression is more appropriate for addressing the research questions posed in our study.  We hope that these arguments clarify our decision to use a linear regression model rather than ANOVA. Nevertheless, we are open to further discussion of any additional concerns or suggestions you may have regarding our methodological approach.

Round 2

Reviewer 4 Report

The authors have addressed to all my concerns.